# An autoregulatory negative feedback loop controls thermomorphogenesis in Arabidopsis

**Sanghwa Lee**, **Ling Zhu**¤, **Enamul Huq***

Department of Molecular Biosciences and The Institute for Cellular and Molecular Biology, The University of Texas at Austin, Austin, Texas, United States of America

¤ Current address: Syngenta Crop protection, LLC-NC, Research Triangle Park, Durham, North Carolina, United States of America
* huq@austin.utexas.edu

**Data Availability Statement:** RNA sequencing data are available from the Gene Expression Omnibus database (accession number GSE158992). All other relevant data are within the paper and its supporting information file.

## Abstract

Plant growth and development are acutely sensitive to high ambient temperature caused in part due to climate change. However, the mechanism of high ambient temperature signaling is not well defined. Here, we show that HECATEs (HEC1 and HEC2), two helix-loop-helix transcription factors, inhibit thermomorphogenesis. While the expression of *HEC1* and *HEC2* is increased and HEC2 protein is stabilized at high ambient temperature, *hec1hec2* double mutant showed exaggerated thermomorphogenesis. Analyses of the four PHYTO-CHROME INTERACTING FACTOR (PIF1, PIF3, PIF4 and PIF5) mutants and overexpression lines showed that they all contribute to promote thermomorphogenesis. Furthermore, genetic analysis showed that *pifQ* is epistatic to *hec1hec2*. HECs and PIFs oppositely control the expression of many genes in response to high ambient temperature. PIFs activate the expression of *HEC*s in response to high ambient temperature. HEC2 in turn interacts with PIF4 both in yeast and *in vivo*. In the absence of HECs, PIF4 binding to its own promoter as well as the target gene promoters was enhanced, indicating that HECs control PIF4 activity via heterodimerization. Overall, these data suggest that PIF4-HEC forms an autoregulatory composite negative feedback loop that controls growth genes to modulate thermomorphogenesis.

## Author summary

Global warming caused by climate change affects all terrestrial life forms including crop yield. Proper response to high ambient temperature is crucial for survival of plants. However, our understanding of how plants respond to high ambient temperature is still rudimentary. Here we show that the HECATE and the PHYTOCHROME INTERACTING FACTORs (PIFs) family of basic helix loop helix proteins antagonistically regulate thermomorphogenesis. PIFs transcriptionally activate the expression of *HEC*s in response to high ambient temperature. HEC2 in turn physically heterodimerizes with PIF4 and inhibit its function. Because PIF4 controls its own expression, this PIF-HEC negative feedback

**Funding:** This work was supported by grants from the National Science Foundation (MCB-2014408) and National Institute of Health (NIH) (GM-114297) to E.H., and Integrative Biology (IB) Research Fellowship grant from the University of Texas at Austin to S.L. The funders had no role in study design, data collection and analysis, decision to publish, or preparation of the manuscript.

**Competing interests:** The authors have declared that no competing interests exist.

loop fine tunes thermomorphogenesis in a temperature-dependent manner. Our data contribute to a deeper understanding of how plants respond to high ambient temperature and are expected to help develop next generation crops that are resilient to global warming.

## Introduction

Temperature is one of the most influential environmental factors affecting all terrestrial life forms. As such, a small increase in ambient temperature due to global warming can have profound impacts on disease resistance, crop yield and ultimately ecological balance [1–3]. High ambient temperature-mediated regulation of plant development is termed thermomorphogenesis, which is characterized by elongated hypocotyl, petiole and root, hyponastic growth, and early flowering [4,5]. These morphological changes help plants adapt to a new climate and complete reproduction.

Plants use diverse mechanisms to perceive high ambient temperature [6,7]. In the model plant *Arabidopsis thaliana*, the red-light photoreceptor phytochrome B (phyB) senses temperature by a process called thermal relaxation, where the active Pfr form of phyB is converted back to the inactive Pr form by high ambient temperature [8,9]. A prion-like domain in EARLY FLOWERING 3 (ELF3) is necessary for the conversion of the soluble active form to an inactive phase-separated form of ELF3 at high ambient temperature [10]. In addition, similar to the bacterial RNA thermometer, an RNA thermoswitch within the 5'-untranslated region of PHYTOCHROME INTERCTING FACTOR 7 (PIF7) acts as a temperature sensor to regulate the translational efficiency of *PIF7* mRNA [11].

The signaling pathway regulating thermomorphogenesis involves multiple factors [4]. Among these, PIF4 acts as a crucial hub transcription factor regulating this process [12]. Both the expression and stability of PIF4 is upregulated by high ambient temperature, which in turn controls downstream genes involved in auxin signaling and biosynthesis to promote hypocotyl elongation [12–16]. Furthermore, several factors have been shown to control either the PIF4 abundance and/or activity to regulate thermomorphogenesis. Among the positive regulators of PIF4, HEMERA (HMR) stabilizes PIF4 to promote thermomorphogenesis [17]. BBX18/23 promotes PIF4 activity through inhibition of ELF3 function [18]. TEOSINTE BRANCHED 1/ CYCLOIDEA/PCF (TCP) transcription factors directly control the expression as well as the activity of PIF4 by physical interaction to promote thermomorphogenesis [19,20]. The cold response regulator CBF1 and MYB30 directly control the expression as well as the stability of PIF4 by inhibiting phyB-PIF4 interaction [21,22]. SHORT HYPOCOTYL UNDER BLUE1 associates with the clock components CCA1/LHY, which directly binds to the *PIF4* promoter and activate *PIF4* transcription [23]. The CONSTITUTIVE PHOTOMORPHOGENIC 1 (COP1)-DEETIOLATED 1 (DET1)-ELONGATED HYPOCOTYL 5 (HY5) module also controls PIF4 both transcriptionally and post- translationally [24–27]. Finally, *SUPPRESSOR OF PHYA-105* (*SPA*) family of genes have been shown to regulate thermomorphogenesis in part via controlling the phyB-PIF4 module in part through its kinase activity [28].

Among the negative regulators of PIF4, the blue-light receptor Cryptochrome 1 (Cry1) directly interacts with PIF4 in a blue light-dependent manner to inhibit thermomorphogenesis [29]. Ultraviolet-B light (UV-B) perceived by the photoreceptor UV RESISTANCE LOCUS 8 (UVR8) attenuates thermomorphogenesis either by repressing the expression of *PIF4* via UVR8-COP1 complex and/or UV-B-mediated stabilization of HFR1, which suppresses PIF4 activity at high ambient temperature [30]. The RNA-binding protein FCA inhibits PIF4

promoter occupancy by direct physical interaction with PIF4 and attenuates thermomorphogenesis [31]. TIMING OF CAB EXPRESSION 1 (TOC1) suppresses thermoresponsive growth in the evening by inhibiting PIF4 activity [32]. EARLY FLOWERING 4 (ELF4) stabilizes ELF3 function, which inhibits PIF4 activity independent of the circadian clock to regulate thermomorphogenesis [10,33,34]. Furthermore, GIGANTEA (GI) represses thermomorphogenesis by stabilizing REPRESSOR OF ga1-3 (RGA), which inhibits PIF4 activity [35]. While these factors control the activity of PIF4 to regulate thermomorphogenesis, here we report a unique autoregulatory mechanism where PIF4 and HECs control their own expression. Moreover, HECs form a composite negative feedback loop with PIF4 that controls growth genes to modulate thermomorphogenesis.

## Results

### HECATEs (HECs) inhibit thermomorphogenesis

*HECATE* (*HEC*) genes were originally reported not only as essential components in the reproductive tissue development, but also as negative regulators of photomorphogenesis [36–38]. In our recent RNA-seq analysis of thermo-regulated gene expression [28], we found that *HECATE 1* and *2* are both induced after high ambient temperature exposure (S1 Fig). Therefore, qPCR was conducted for independent verification of the RNA-seq result using 6-day-old seedlings grown at 22˚C or transferred to 28˚C for additional 24 hours under continuous white light. As expected, the transcript levels of both *HEC1* and *HEC2* were increased at high ambient temperature (Fig 1A). To investigate the potential temperature regulation of HEC2 at the post-translational level, we examined HEC2 protein level at 22˚C and 28˚C using *HEC2-GFP* overexpression line driven by the constitutively active cauliflower mosaic virus (CaMV) 35S promoter. The result shows that HEC2 protein level was accumulated at high ambient temperature without alteration of the transcript level (Figs 1B and S2), indicating that high ambient temperature stabilizes HEC2 post-translationally. To investigate whether HEC1 and HEC2 play any role in regulating thermomorphogenesis, we examined the phenotype of *hec1hec2* double mutant and *35S:HEC2-GFP* overexpression line at high ambient temperature. Intriguingly, *hec1hec2* double mutant showed longer hypocotyl and petiole length, whereas *35S: HEC2-GFP* showed shorter hypocotyl and petiole length compared to the wild-type at high ambient temperature (Figs 1C, 1D and S3A). Furthermore, *hec1 hec2* mutant showed early flowering while *35S:HEC2-GFP* and Col-0 remained the same at high ambient temperature (S3B Fig). Overall, these data show that *HEC*s are induced at high ambient temperature both transcriptionally and post-translationally, and act as negative regulators of thermomorphogenesis.

### PIF family members contribute to thermomorphogenesis

Among all PIFs, only PIF4 and PIF7 have been shown to regulate thermomorphogenesis [11,12,39]. Because PIF family members interact with HECs [37], we examined whether other PIFs also contribute to regulating thermomorphogenesis. Therefore, we tested high ambient temperature mediated hypocotyl and petiole elongation phenotypes using the higher order *pif* mutants and 35S:*PIF*-overexpression lines (*PIF1*, *PIF3*, *PIF4* and *PIF5*). The results show that *pif* single, double, triple and quadruple mutants display attenuated or no response at 28˚C compared to 22˚C, and all four *PIF*-overexpression lines display elongated hypocotyl at high ambient temperature (Figs 2A–2C and S4A and S4B). Among *pif* single mutants, *pif4* displayed the strongest attenuated response as previously shown [12]; however, other PIFs, such as PIF1, PIF3, and PIF5 also contribute to regulating thermomorphogenesis at high ambient

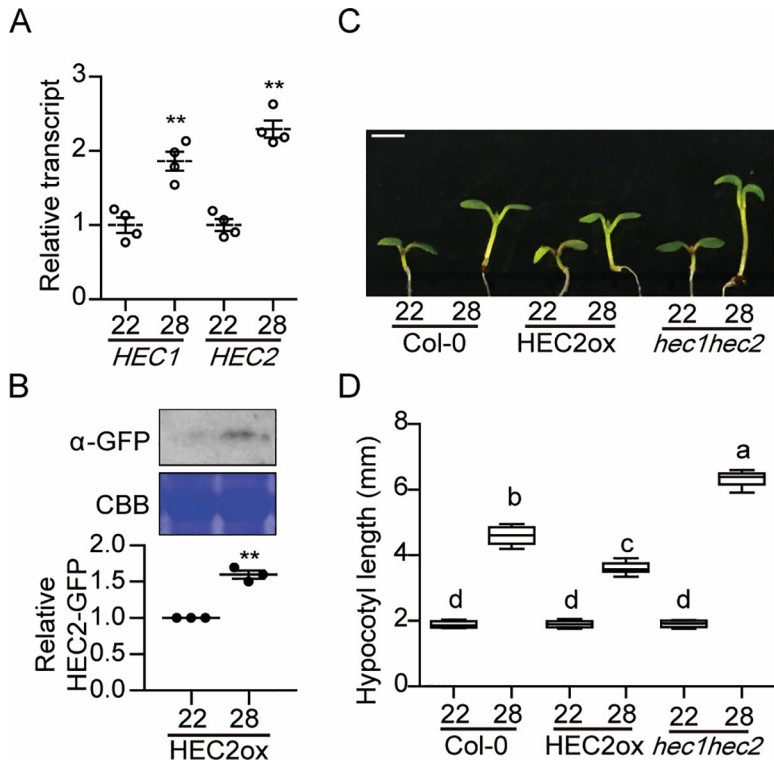

**Fig 1. *HEC*s inhibit thermomorphogenesis.** (A) The expression of *HEC1* and *HEC2* is upregulated by high ambient temperature. RT-qPCR samples were from Col-0 whole seedlings grown for 5 days at 22˚C and then either kept at 22˚C or transferred to 28˚C for 24 hours. Four biological repeats were performed. Relative gene expression levels were normalized using *ACT7*. (B) Western blot shows the level of HEC2-GFP from *35S:HEC2-GFP* whole seedlings grown for 5 days at 22˚C and either kept at 22˚C or transferred to 28˚C for 4 hours. Coomassie staining was used as a loading control. The experiments were repeated three times with similar results as indicated in the bar graph in lower panel. (C) Photograph showing seedling phenotypes of *35S:HEC2-GFP* and *hec1hec2* in normal and high ambient temperature. Seedlings were grown for two days in continuous white light at 22˚C and then either kept at 22˚C or transferred to 28˚C for additional four days before being photographed. The scale bar represents 5 mm. (D) Box plot shows the hypocotyl lengths grown under the conditions described in (C). More than 10 seedlings were measured for each experiment and was repeated 3 times. The letters a-d indicate statistically significant differences based on one-way ANOVA analysis with Tukey's HSD test. Tukey's box plot was used with median as a center value. The experiments were repeated more than three times with similar results.

temperature (Fig 2A–2C). Taken together, these data suggest that *PIF* family members collectively contribute to regulating thermomorphogenesis.

Our previous report showed that stabilized PIF4 is correlated with destabilized phyB at high ambient temperature [28]. Therefore, we conducted immunoblot assays using 5-day-old seedlings of *pif4*, *pifQ* and *PIF*-overexpression lines with or without 28˚C treatment for 4 hours. Consistent with their phenotypes, phyB level was up-regulated in the *pif4* and more robustly in *pifQ* mutant background at 22˚C (Fig 2D and 2E). However, in response to high ambient temperature, phyB level was reduced in both *pif4* and *pifQ* backgrounds at 28˚C (Fig 2D and 2E). Conversely, all four PIFs were stabilized at high ambient temperature (S4C and S4D Fig). In contrast, phyB levels were decreased in the wild-type as well as in four *PIF*-overexpression lines in response to high ambient temperature (S4E and S4F Fig). Thus, PIFs and phyB are inversely regulated at high ambient temperature as previously observed under red light conditions [40].

To investigate whether the altered protein levels of phyB and PIFs are caused by transcriptional regulation, we conducted qPCR to check the transcript levels of *phyB* and *PIFs* in *PIF-*

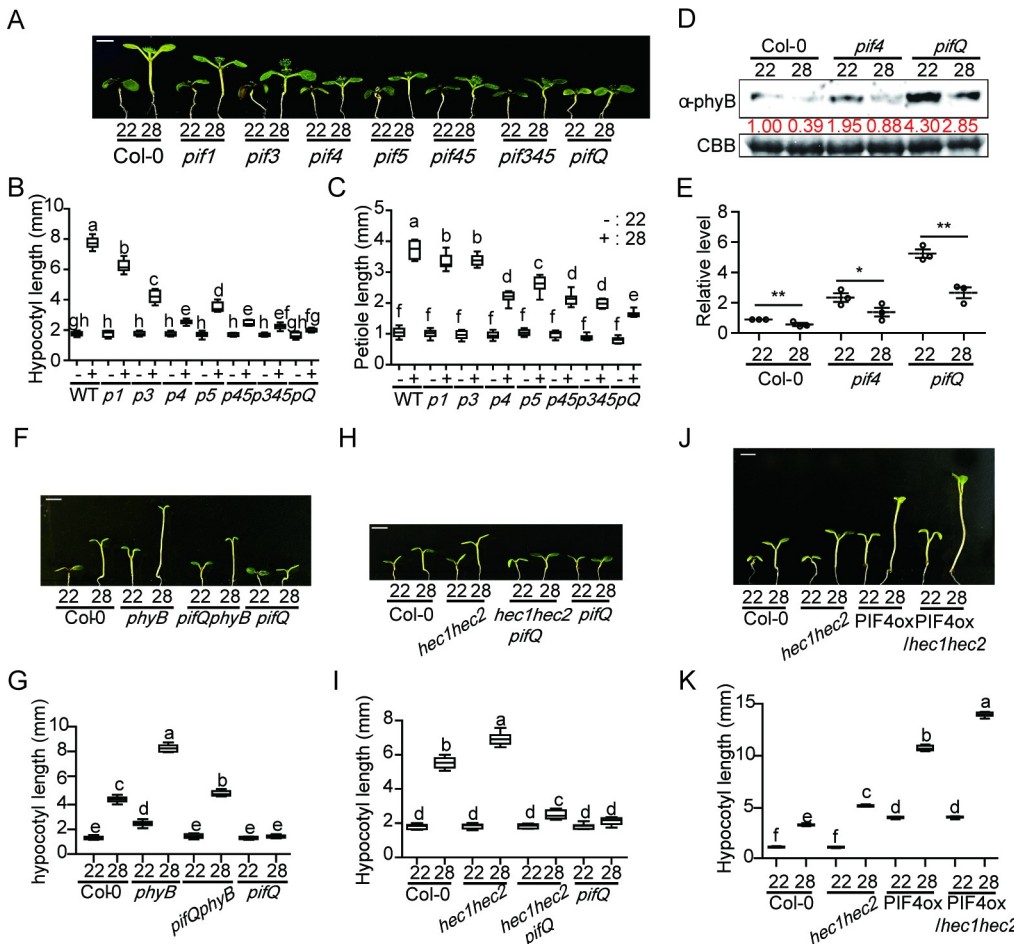

**Fig 2. Genetic relationship of PIFs and HECs in thermomorphogenesis.** (A) Photograph showing seedling phenotypes of high order *pif*s at normal and high ambient temperature conditions. Seedlings were grown for two days under continuous white light at 22˚C and then either kept at 22˚C or transferred to 28˚C for additional six days before being photographed. The scale bar represents 5 mm. (B and C) Dot plot shows the hypocotyl and petiole lengths of seedlings grown under conditions described in (A). (D) Western blot shows the level of phyB from whole seedlings grown for 5 days in 22˚C and either kept at 22˚C or transferred to 28˚C for 4 hours. Coomassie staining was used as a control. Red number indicates the quantitation value from anti-phyB divided by the control. (E) Bar graph shows the relative amount of phyB (n = 3). Asterisks indicate statistically significant difference using Student's t-test; $^{*}$p < 0.05 and $^{**}$p < 0.01. The experiments were repeated three times with similar results. (F) Photograph shows the seedling phenotypes of *phyB*, *pifQphyB* and *pifQ* at normal and high ambient temperature. Seedlings were grown as described in (A). (G) Box plot shows the hypocotyl lengths for seedlings described in (F). (H) Photograph showing seedling phenotypes of WT, *hec1hec2*, *hec1hec2 pifQ* and *pifQ* in normal and high ambient temperature. More than 10 seedlings were measured for each experiment and was repeated 3 times. (I) Box plot shows the hypocotyl length of seedlings grown under conditions described in (A). The letters a-g indicate statistically significant differences based on one-way ANOVA analysis with Tukey's HSD test. Tukey's box plot was used with median as a center value. (J) Photograph showing seedling phenotypes of WT, *hec1hec2*, *35S::PIF4-myc* in either Col-0 or *hec1hec2* background at high ambient temperature. Conditions are the same as described in (A). (K) Box plot shows the hypocotyl lengths of seedlings grown under conditions described in (J). More than 10 seedlings were measured for each experiment and was repeated 3 times. The letters a-f indicate statistically significant differences based on one-way ANOVA analysis with Tukey's HSD test. Tukey's box plot was used with median as a center value.

overexpression lines with or without 28˚C treatment for 4 hours. As expected, the transcript level of *PHYB* and *PIF*-overexpression lines did not alter in wild-type and *PIF*-overexpression lines, respectively after high ambient temperature treatment (S5A and S5B Fig). However, examination of the expression level of native *PIF*s showed that all four *PIF*s are

transcriptionally up-regulated with the *PIF4* being at the highest level in response to high ambient temperature (S6 Fig). Thus, PIFs are up-regulated both transcriptionally and post-translationally, while phyB is down-regulated only post-translationally in response to high ambient temperature. The altered phyB-PIF ratio contributes to the regulation of thermomorphogenesis.

Finally, we also examined the phenotype of *phyB pifQ* to test the genetic relationships between *phyB* and *PIF*s at high ambient temperature. Interestingly, *phyB pifQ* showed an intermediate phenotype in response to high ambient temperature (Fig 2F and 2G), implying other factors (e.g., PIF7 and others) might be involved in this process. Taken together, these data suggest that as a temperature sensor, phyB is acting as a master regulator, while PIFs are acting positively downstream of phyB in promoting thermormoprhogenesis.

## Genetic relationships between HECs and PIFs

To examine the genetic relationships between *HECs* and *PIFs*, *hec1 hec2 pifQ* hexuple mutant and *PIF4* overexpression line in *hec1 hec2* background were generated. At high ambient temperature, *hec1 hec2 pifQ* showed strongly reduced hypocotyl and petiole elongation phenotype compared to parental genotypes (Figs 2H, 2I and S7A). The hexuple mutant phenotypes are almost similar to that of *pifQ*. In addition, the thermo-induced flowering time of *hec1 hec2* mutant was inhibited in *hec1 hec2 pifQ* (S7B Fig), indicating that *pifQ* is epistatic to *hec1 hec2* in response to high ambient temperature. Moreover, the elongated phenotype of the *PIF4* overexpression line was exaggerated in the *hec1 hec2* background (Fig 2J and 2K), suggesting that HECs genetically attenuate PIF4 function. Overall, these data suggest that *HECs* and *PIFs* are regulating thermomorphogenesis in an inter-dependent manner.

## *HEC*s alter global gene expression at high ambient temperature

High ambient temperature induces global gene expressions to promote thermomorphogenesis [9,18,28]. To examine whether *HEC*s control thermo-induced gene expression, RNA-seq was conducted using wild-type Col-0, *hec1 hec2* double mutant and *pifQ* at both 22˚C and 28˚C (Fig 3). The results show that 2008, 2209 and 1837 genes were differentially expressed in Col-0, *hec1 hec2* and *pifQ*, respectively in response to high ambient temperature (Fig 3A). Among the three genotypes, 735 genes were co-regulated. Gene Ontology (GO) analysis revealed that 735 temperature-dependent genes are mainly involved in pigment biosynthetic pathways (S8 Fig). Examination of the HEC- and PIF-dependent genes display opposite expression patterns for many of these genes (S9A and S9B Fig). GO analysis of the HEC-dependent genes shows an enrichment of the genes involved in leucine and glucosinolate biosynthesis, plant cell-wall organization, organ morphogenesis and others (Fig 3B), while the PIF-dependent genes show that they are involved in auxin signaling, translational processes and leaf morphogenesis (Fig 3C). These data reinforce previous conclusions that PIFs regulate auxin signaling to modulate organ morphogenesis in response to high ambient temperature. Furthermore, 2008 DEGs in wild-type displayed distinct patterns in *hec1 hec2* and *pifQ* as shown in the heatmap analysis (Fig 3D). Overall, these data suggest that HECs and PIFs control the expression of a large number of genes to regulate thermomorphogenesis.

To independently verify the RNA-seq data, qPCR assays were performed to examine the transcript level of several thermo-responsive marker genes such as *YUC8* and *IAA29* (Fig 3E and 3F). Consistent with the RNA-seq data, the transcript level of these genes was upregulated in response to high ambient temperature and correlated with thermomorphogensis as previously shown [15,29,30]. Interestingly, the transcript level of *LONG HYPOCOTYL IN FAR-RED 1* (*HFR1*) and *PIF3-LIKE 1* (*PIL1*), which are known inhibitors of PIFs in

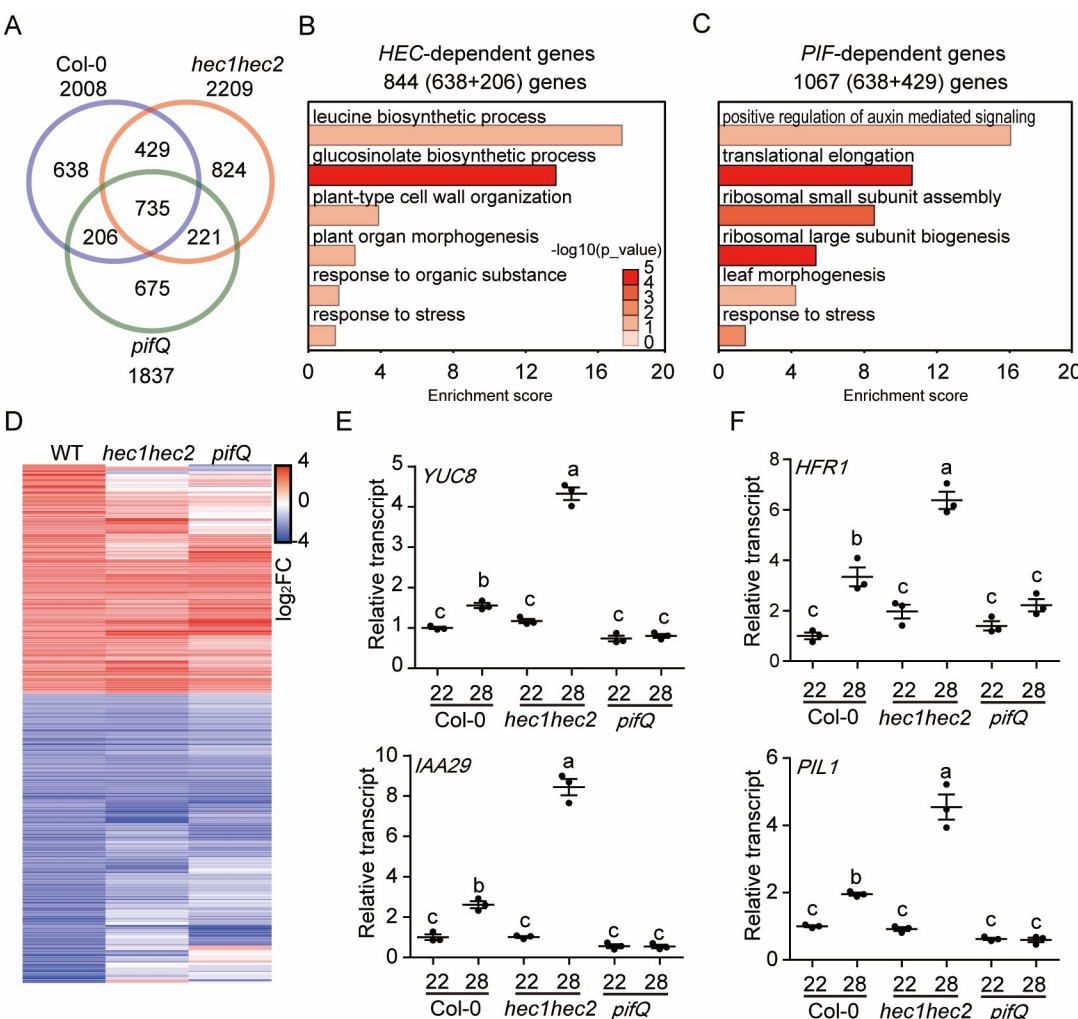

**Fig 3. _HEC_s regulate global gene expression at high ambient temperature.** (A) Venn diagrams show differentially expressed genes (DEGs) in wild-type (WT) vs _hec1hec2_ vs _pifQ_ mutant at high ambient temperature. Six-day-old white light-grown seedlings were transferred to 22°C or 28°C for additional 24 hours and total RNA was extracted from three biological replicates for RNA-seq analyses. (B-C) Gene Ontology (GO) analysis of _HEC_-dependent 844 genes (B) and _PIF_-dependent 1067 genes (C). (D) Hierarchical clustering from 2008 DEGs from WT show distinct pattern in _hec1hec2_ and _pifQ_ mutant at high ambient temperature. (E-F) RT-qPCR analysis using growth genes (E), and PIF inhibitor genes (F). RT-qPCR samples were from whole seedlings grown for 6 days at 22°C and then either kept at 22°C or transferred to 28°C for 24 hours. Three biological repeats were performed. Relative gene expression levels were normalized using the expression levels of _ACT7_ for RT-qPCR. The letters a-c indicate statistically significant differences based on one-way ANOVA analysis with Tukey's HSD test.

photomorphogenesis, thermomorphogenesis and shade avoidance responses [41–46], were up-regulated in the absence of _HEC_s (Fig 3F). These data suggest that similar to _HEC1_ and _HEC2_, genes encoding the inhibitors of PIFs are also induced under high ambient temperature responses and perhaps participate in the negative feedback loop to regulate PIF level and activity at high ambient temperature. Taken together, these data also suggest that _HEC_s play an important role in thermomorphogenesis via transcriptional regulation.

## HECs alter phyB-PIF4 level at high ambient temperature

It has been shown that the protein level of PIF4 is crucial in regulating thermomorphogenesis due to its transcriptional regulation of growth genes via direct promoter binding [13,15].

Furthermore, we recently showed that the thermosensor phyB and the hub transcription factor PIF4 levels are oppositely regulated where low phyB level is correlated with high PIF4 level and vice versa at high ambient temperature [28]. To investigate whether phyB as well as PIF4 levels are controlled by HECs, Western blot analyses were conducted using *hec1 hec2* mutant and *35S:HEC2-GFP* seedlings grown at 22˚C for 5 days and kept in 22˚C or transferred to 28˚C for 4 hours. Consistent with the phenotype (Fig 1C and 1D), *hec1 hec2* mutant displayed reduced level of phyB at both 22˚C and 28˚C compared to that of wild-type (Fig 4A and 4B). The *35S:HEC2-GFP* transgenic plant showed higher level of phyB compared to that of the wild-type at both temperatures, indicating that HECs contribute to stabilize phyB level. In contrast, PIF4 level is strongly stabilized in the *35S:HEC2-GFP* transgenic plant at both temperature and modestly stabilized in the *hec1 hec2* mutant compared to wild type at high ambient temperature (Fig 4A and 4B). While the phyB level in *hec* mutants correlates with the thermomorphogenesis phenotype, the PIF4 level did not correlate with the phenotype in *hec* mutants. These data suggest that HECs not only alter the phyB-PIF4 level at high ambient temperature, but also might regulate the activity of PIF4 to modulate thermomorphogenesis.

## HEC2 regulates thermomorphogenesis in part by heterodimerizing with PIF4

HECs are known to interact with PIFs and regulate PIF1-mediated seed germination in a heterodimerization-dependent manner [37]. To examine if HECs regulate thermomorphogenesis through PIF4 in a heterodimerization-dependent manner, we tested the interaction of PIF4 with wild type HEC2 and a dimerization-defective mutant (mHEC2) both in yeast and in transgenic plants. Results show that PIF4 heterodimerizes with wild type HEC2 in yeast-two-hybrid assay, but not the dimerization-defective mutant HEC2 (Fig 5A). *In vivo* coimmunoprecipitation assay also showed that PIF4 interacts with the wild type HEC2 but not the mutant HEC2 (Fig 5B). Thus, HEC2 and possibly other HECs might regulate thermomorphogenesis by direct heterodimerization with PIF4.

To test the above hypothesis more directly, we examined the phenotype of *35S:mHEC2-GFP* plant compared to *35S:HEC2-GFP* transgenic plant. The results show that the wild type *HEC2* overexpression line (*35S:HEC2-GFP*) attenuates thermomorphogenesis, while the *35S:mHEC2-GFP* plant display hypocotyl lengths similar to that of wild-type (Fig 5C and 5D), suggesting that heterodimerization of HEC2 with PIF4 is necessary for regulating thermomorphogenesis. Both *35S:HEC2-GFP* and *35S:mHEC2-GFP* did not affect flowering time at high ambient temperature (S10 Fig). We also performed Western blot analysis using phyB and PIF4 antibodies to test whether HEC2 regulates phyB and PIF4 level during thermomorphogenesis. Results show that the wild type HEC2 stabilized both phyB and PIF4 level in response to high ambient temperature, while the *35S:mHEC2-GFP* plant showed similar levels of phyB and PIF4 to that of the wild-type (Fig 5E and 5F). Moreover, PIF4-myc is less abundant in the *hec1 hec2* double mutant background compared to wild type at 22˚C and 28˚C (Fig 5G). To test whether the reduced level of PIF4 in *hec1 hec2* background is due to the 26S proteasome-mediated degradation, we treated seedlings with the proteasome inhibitor bortezomib before Western blot was performed. Results show that PIF4-myc level was stabilized in the *hec1 hec2* mutant and became similar to that of wild-type (S11 Fig), suggesting that HEC-PIF4 heterodimerization inhibits the 26S proteasome-mediated degradation of PIF4. These data are consistent with HEC2-mediated stabilization of PIF1 [37]. Overall, these data suggest that HEC2 stabilizes both phyB and PIF4, but might inhibit PIF4 function by heterodimerization during thermomorphogenesis.

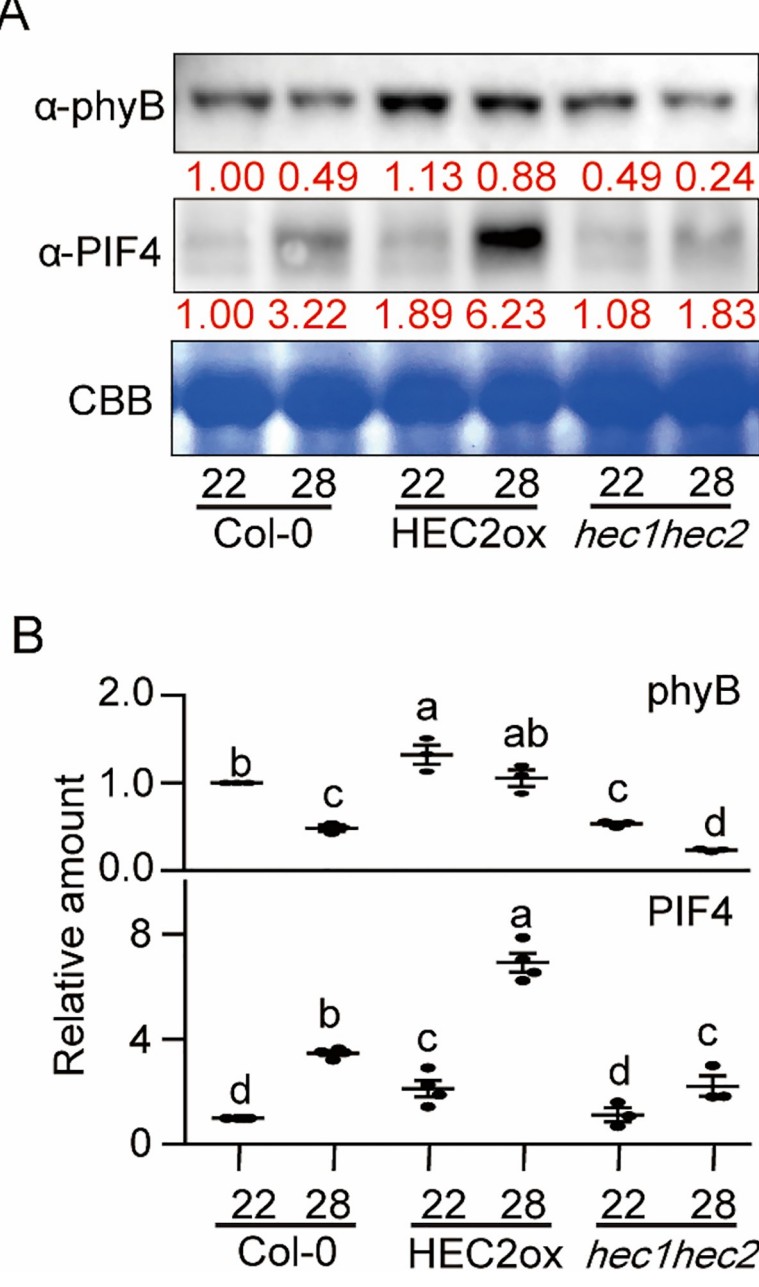

**Fig 4. HECs alter phyB-PIF4 level at high ambient temperature.** (A) Western blot shows the level of phyB or PIF4 in WT, *35S:HEC2-GFP* and *hec1hec2* mutant. Whole seedlings were grown for 5 days at 22˚C and either kept at 22˚C or transferred to 28˚C for 4 hours. Coomassie staining was used as a loading control. Red number indicates the quantitation value from anti-phyB or anti-PIF4 levels divided by Coomassie blue staining intensity. (B) Dot plot shows the relative amount of phyB (n = 3) or PIF4 (n = 3 or 4). Asterisks indicate statistically significant differences using Student's t-test; **p < 0.01.

## HECs inhibit PIF4 promoter binding

PIF4 binds to the promoters of growth genes and promotes their transcription [13,15]. Since HEC2-PIF4 interaction was crucial for inhibiting thermomorphogenesis (Fig 5), we hypothesized that heterodimerization of HEC2-PIF4 could suppress the binding of PIF4 to the target

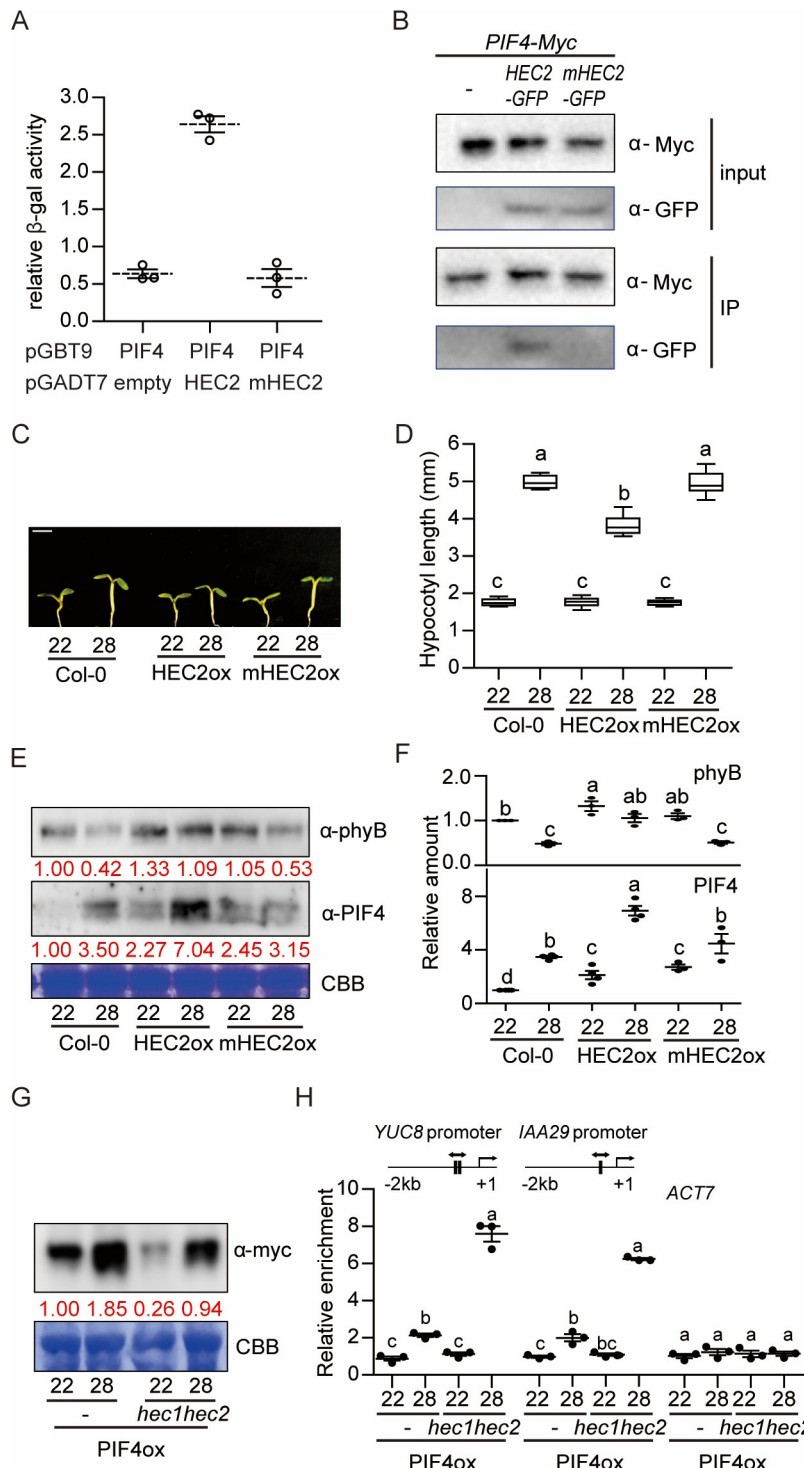

**Fig 5. PIF4-HEC2 heterodimerization is crucial for HEC2 function in thermomorphogenesis.** (A) HEC2 interacts with PIF4. Yeast two hybrid assay using PIF4 a bait and HEC2 or mHEC2 as preys. The error bars represent standard deviation. Three biological replicates were used in this study. (B) HEC2 interacts with PIF4 *in vivo*. *In vivo* coimmunoprecipitation assay was performed using *35S:PIF4-myc* with *35S:HEC2-GFP* or *mHEC2-GFP* crossed line. 35S:PIF4-myc was used as a control. (C) Photograph showing seedling phenotypes of WT, *35S:HEC2-GFP* and *35S: mHEC2-GFP* grown at normal and high ambient temperature conditions. Seedlings were grown for two days under continuous white light at 22˚C and then either kept at 22˚C or transferred to 28˚C for additional four days before being photographed. The scale bar represents 5 mm. (D) Box plot shows the hypocotyl length grown under conditions

described in (C). More than 10 seedlings were measured for each experiment and was repeated 3 times. The letters a-c indicate statistically significant differences between means of hypocotyl lengths (P<0.05) based on one-way ANOVA analysis with Tukey's HSD test. Tukey's box plot was used with median as a center value. n.s. stands for statistically not significant. (E) Western blots show the level of phyB and PIF4 in WT, *35S:HEC2-GFP* and *35S:mHEC2-GFP*. Coomassie staining was used as a control. Red number indicates the quantitation value from anti-phyB or anti-PIF4 detection divided by Coomassie blue staining intensity. (F) Dot plot shows the relative amount of phyB (top, n = 3) or PIF4 (bottom, n = 3). (G) Western blot shows the level of PIF4-myc from *35S::PIF4-myc* in either Col-0 or *hec1hec2* background. Seedlings were grown for 5 days at 22˚C and either kept at 22˚C or transferred to 28˚C for 4 hours. Coomassie staining was used as a control. Red numbers indicate the anti-myc intensity divided from Coomassie control. (H) ChIP-qPCR analysis using promoter regions of *YUC8* and *IAA29*. Simple genomic region scheme is described with PIF4 binding motif in vertical line. *ACT7* was used as a control. ChIP-qPCR samples were from whole seedlings grown for 6 days at 22˚C and then either kept at 22˚C or transferred to 28˚C for 24 hours. Three biological repeats were performed. Relative abundance was normalized from % of input value of *35S::PIF4-myc* at 22˚C for ChIP-qPCR.

promoters. To test this hypothesis, we performed *in vitro* DNA binding of PIF4 in the absence and presence of HEC2. Result shows that PIF4 robustly binds to DNA even in the presence of GST control. However, addition of HEC2 along with PIF4 strongly inhibits PIF4 binding to DNA (S12 Fig). To substantiate these *in vitro* data, we generated *35S:PIF4-Myc* in *hec1 hec2* mutant and compared the promoter binding to that of the *35S:PIF4-Myc* in Col-0 background. Chromatin Immunoprecipitation (ChIP) assays were performed using the *35S:PIF4-Myc* in Col-0 and *hec1 hec2* mutant backgrounds at both 22˚C and 28˚C. The results show that even though PIF4 is less abundant in the *hec1 hec2* background compared to wild type (Figs 5G and S11), the enrichment of PIF4 on the direct target gene promoters such as *YUC8* and *IAA29* was strongly higher in the *hec1 hec2* mutant compared to Col-0 background (Fig 5H). These data are consistent with the expression level of these genes (Fig 3E), as well as the thermo-induced hypocotyl elongation phenotype of the *35S:PIF4-Myc/hec1 hec2* compared to *35S:PIF4-Myc* (Fig 2J and 2K).

Previously, PIF1, PIF3, PIF4 and PIF5 have been shown to directly regulate the expression of *HECs* in darkness [37]. To examine if PIFs also regulate the expression of *HECs* under high ambient temperature, we performed RT-qPCR for *HEC1* and *HEC2* in wild type, *pif4* and *pifQ* grown at 22˚C and 28˚C. Results show that the thermo-induced expression of both *HEC1* and *HEC2* is strongly down-regulated in the *pif4* and almost eliminated in the *pifQ* background (Fig 6A). Consistent with the expression level, PIF4 binding to the *HEC1* and *HEC2* promoters was also enhanced under high ambient temperature (Fig 6C). In contrast, the thermo-induced expression of *PIF4* is strongly up-regulated in the *hec1 hec2* background compared to the wild type (Figs 6B and S13). Strikingly, the binding of PIF4 on its own promoter was also strongly enhanced in the *hec1 hec2* background compared to the wild type (Fig 6D). Thus, PIF4 autoregulates itself as well as the expression of *HECs* under high ambient temperature. Conversely, HECs also regulate the expression and the promoter occupancy of PIF4 to fine tune thermomorphogenesis.

## Discussion

High ambient temperature due to global warming has a far-reaching impact on food security, ecological balance, and planet sustainability [1–3,47]. Fundamental understanding of how plant growth and development is regulated by high ambient temperature will play a vital role in developing crop plants resilient to climate change. In this study, we provide multiple evidence in support of a new role for the HEC family of bHLH transcription factors to fine tune thermomorphogenesis. First, the expression of *HEC1* and *HEC2* is upregulated and HEC2-GFP is post-translationally stabilized in response to high ambient temperature (Figs 1A, 1B, and S1). Second, *hec1 hec2* double mutant display elongated hypocotyl and petiole

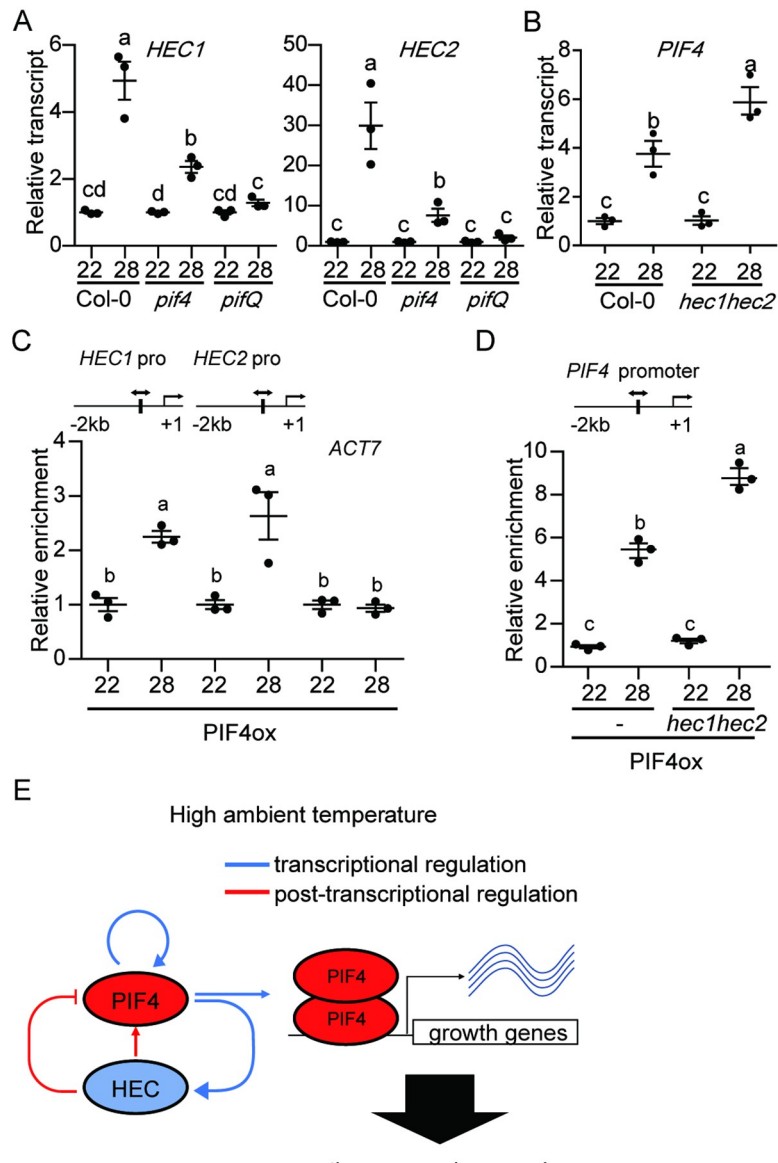

**Fig 6. An autoregulatory negative feedback module controls thermomorphogenesis.** (A) PIFs activate the expression of *HEC1* and *HEC2*. RT-qPCR analysis of *HEC1* and *HEC2* gene expression using Col-0, *pif4* and *pifQ* mutant. (B) HECs inhibit the expression of *PIF4* under high ambient temperature. RT-qPCR analysis of *PIF4* using Col-0 and *hec1hec2* mutant. Relative gene expression levels were normalized using expression levels of *ACT7*. (C) PIF4 directly activates the expression of *HEC*s. ChIP-qPCR analysis of PIF4 binding to the *HEC1* and *HEC2* promoter region using 35S:PIF4-myc in Col-0 background. *ACT7* was used for the control. (D) PIF4 activates its own expression in a HEC-dependent manner. ChIP-qPCR analysis of *PIF4* promoter region using 35S:PIF4-myc in Col-0 and *hec1hec2* backgrounds. For both RT-qPCR and ChIP-qPCR samples, whole seedlings grown for 5 days at 22˚C and then either kept at 22˚C or transferred to 28˚C for 4 hours. Three biological repeats were performed. The letters a-c indicate statistically significant differences based on one-way ANOVA analysis with Tukey's HSD test. Simple genomic region scheme is described with PIF4 binding motif in vertical line. (D) Simplified model shows the role of an autoregulatory module controlling thermomorphogenesis. At high ambient temperature, PIF4 activates the expression of both *PIF4* and *HEC*s by transcriptional regulation. HECs heterodimerize with PIF4 and inhibit PIF4 function which regulates growth genes at transcriptional level.

length, and early flowering (Figs 1C, 1D, and S3), hallmark phenotypes of thermomorphogenesis. In contrast, *HEC2-GFP* overexpression line displays attenuated thermomorphogenesis. Third, HECs regulate the expression of a large number of genes in response to high ambient temperature (Fig 3). Fourth, *hec1 hec2* enhances the exaggerated phenotype of *PIF4* overexpression line under high ambient temperature (Fig 2J and 2K). These data firmly establish HECs as negative regulators of thermomorphogenesis.

Previously, multiple negative regulators have been described in thermomorphogenesis pathway [4]. These include Cry1, ELF3, GI-DELLA module, TOC1, and FCA [29,31–33,35,48]. All of these factors directly interact with the central hub transcription factor, PIF4 and inhibit its function. By contrast, being the same HLH family of transcription factors, HECs heterodimerize with PIF4 and other PIFs, and produce a non-functional PIF-HEC heterodimer that is unable to bind to DNA (Figs 5, 6, and S12) (37). In addition, HECs also stabilize PIF4 and destabilize the thermo sensor phyB (Fig 4A and 4B). Thus, HECs regulate thermomorphogenesis by controlling the activity of PIFs in a heterodimerization-dependent manner. This behavior is reflected in the global gene expression analyses where HECs and PIFs regulate the expression of a large number of genes oppositely (Figs 3 and S9). Consistently, *hec1 hec2 pifQ* hexuple mutant displays similar phenotype as *pifQ*, suggesting PIFs are epistatic to HECs in regulating thermomorphogenesis (Fig 2H and 2I).

Downstream of the thermo sensor phyB, PIF4 has been shown to play central role in regulating thermomorphogenesis [12]. PIF7 also plays important roles in two ways: first, along with PIF4, PIF7 contributes to regulating thermomorphogenesis [39]; and second, an RNA thermoswitch at the 5'untranslated region of *PIF7* mRNA plays a sensory role in perceiving temperature and regulating the translational efficiency of *PIF7* mRNA [11]. Our data show that not only PIF4 and PIF7, other PIFs also contribute to regulating thermomorphogenesis (Figs 2 and S4). The expression and post-translational stability of PIF1, PIF3 and PIF5 are also upregulated by high ambient temperature (S4 and S6 Figs). Overexpression of *PIF1*, *PIF3* and *PIF5* displays longer hypocotyl compared with wild type (S4A and S4B Fig), while the *pif* single, double, triple, and quadruple mutants display an additive role in regulating thermomorphogenesis (Fig 2A–2C). Moreover, *pifQ phyB* quintuple mutant displays an intermediate phenotype between *phyB* and *pifQ* (Fig 2F and 2G), suggesting that other factors including other PIFs might contribute to regulating thermomorphogenesis.

Our data also show that PIF4 regulates its own expression by directly binding to its own promoter as recently described [49]. Thus, in response to high ambient temperature, PIF4 is stabilized and the stable PIF4 activates its own expression as well as other target growth genes (Figs 5H, 6B, 6D and S12) [4]. In addition, PIFs activate the expression of *HEC*s by directly binding to the *HEC* promoters (Fig 6A and 6C) [37]. Under high ambient temperature, PIF4 plays a major role in this regulation in concert with other PIFs (Fig 6A). By contrast, HECs physically interact with PIF4 and inhibit the promoter occupancy to control its own transcription as well as other downstream target genes (Figs 5, 6B, 6D and S12). Consistently, the expression of *PIF4* is strongly enhanced in *hec1 hec2* background in response to high ambient temperature (Figs 6B and S13). In addition, HECs also interact with other PIFs (PIF1, PIF3 and PIF5) and inhibit their promoter occupancy as previously shown [37]. However, these PIFs do not autoregulate themselves, and HECs only inhibit their target gene expression [37]. The HEC function is analogous to the HFR1 function in inhibiting PIFs activity by heterodimerization [42,45,50]. Similar to HECs, the expression and stability of HFR1 are also up-regulated by high ambient temperature [43]. However, the regulation of HFR1 is blue-light-dependent, whereas HECs are regulated under white light conditions. Further studies are necessary to dissect which monochromatic light pathway controls the expression and stability of HECs.

In summary, HECs and PIFs are forming an autoregulatory composite negative feedback loop (Fig 6D), where PIFs transcriptionally activate the expression of *HEC*s in response to high ambient temperature. HECs in turn physically heterodimerize with PIFs and stabilize them post-translationally. However, even though PIFs are stabilized by HECs, PIF activity is inhibited by HECs by direct heterodimerization. Because PIF4 controls its own expression, this PIF-HEC negative feedback loop fine tunes thermomorphogenesis in a temperature-dependent manner.

## Materials and methods

### Plant materials, growth conditions and phenotypic analyses

In this study, mutants and transgenic lines were generated from Col-0 ecotype of *Arabidopsis thaliana*. Seeds were surface-sterilized and plated on Murashige and Skoog (MS) medium without sucrose. Seeds were stratified for 3 days at 4°C in dark, and the plates were placed at 22°C for 2 days and then transferred to 22°C or 28°C for additional 4 days. Hypocotyl and petiole lengths were measured using ImageJ software (n>10) and statistically analyzed using one-way ANOVA analysis with Tukey's HSD test. For flowering time measurement, plants were grown under long day conditions (16L:8D) and then the leaf number was counted at bolting for >10 plants.

### Protein extraction and Western blot analyses

Total protein extracts were made from 50 seedlings for each sample using 50 μL urea extraction buffer [8 M urea, 0.35 M Tris-Cl pH 7.5, and 1× protease inhibitor cocktail]. Subsequently, 6X SDS loading buffer was added to the samples and boiled for 5 min. Supernatant from the samples was loaded into SDS-PAGE gels after centrifugation at 16,000 g for 15 min. PVDF membrane (EMD Millipore, Burlington, MA) were used for transfer, and Western blots were detected using anti-myc (Cell Signaling Technologies, Danvers, MA), anti-GFP (Abiocode, Agoura Hills, CA), anti-phyB, anti-PIF4 (AgriSera AB, Vännäs, Sweden) or anti-RPT5 (Abiocode, Agoura Hills, CA) antibodies. Coomassie blue staining and/or anti-RPT5 were used for the loading control.

### RNA extraction, cDNA synthesis, and qRT-PCR

RNA extraction was performed as previously described [28]. Briefly, 6-day-old white light-grown seedlings were used with three independent biological replicates (n = 3). Seeds were kept in 22°C under continuous white light for 6 days. After 6 days, seedlings were either kept at 22°C or transferred to 28°C for additional 24 hours under continuous white light. Plant RNA purification kit (Sigma-Aldrich Co., St. Louis, MO) were used for total RNA extraction according to the manufacturer's protocols. For cDNA synthesis, 1 mg of total RNA was used for reverse transcription with M-MLV Reverse Transcriptase (Thermofischer Scientific Inc., Waltham, MA). SYBR Green PCR master mix (Thermofischer Scientific Inc., Waltham, MA) and gene-specific oligonucleotides were used to conduct qPCR analyses using primers shown in S1 Table. Finally, relative transcription level was calculated using $2^{\Delta Ct}$ using *ACT7* normalization.

### RNA-seq analyses

3'Tag-Seq method was used in this study for RNA-seq analysis [51]. FastQC was used to examine raw read quality ([www.bioinformatics.babraham.ac.uk/projects/fastqc/](http://www.bioinformatics.babraham.ac.uk/projects/fastqc/)). The raw reads were aligned to the Arabidopsis genome with Bowtie2 [52] and TopHat [53]. The annotation

of the Arabidopsis genome was from TAIR10 (www.arabidopsis.org/). Read count data were obtained using HTseq [54] (htseq.readthedocs.io/en/master/). Differentially expressed genes in WT/*hec1 hec2*/*pifQ* were identified using the EdgeR [55]. Cutoff and adjusted P value (FDR) for the differential gene expression were defined ≥2-fold and ≤0.05 respectively. Venn diagrams were generated using the website (http://bioinformatics.psb.ugent.be/webtools/Venn/) and Heatmap was generated using Morpheus (https://software.broadinstitute.org/morpheus/). For the heatmap analysis, we used the hierarchical clustering with one minus cosine similarity metric combined with average linkage method. Also, GO enrichment analyses were performed using (http://geneontology.org). GO bar graphs were generated based on the result of the significant enriched terms with the lowest P value and FDR (≤0.05) in GO terms. Raw data and processed data for RNAseq in Col-0, *hec1 hec2* and *pifQ* can be accessed from the Gene Expression Omnibus database under accession number GSE158992.

## Yeast two-hybrid analyses

Cloning of *HEC2* and *mHEC2* to pGADT7 and *PIF4* into pGBT9 was described previously [37]. Different combinations of plasmids were introduced into yeast strain Y187 and selected on -Leu, -Trp minimal synthetic medium. β-galactosidase assay was performed according to the manufacturer's protocol (Matchmaker Two-Hybrid System; Takara Bio, Mountain View, CA, https://www.takarabio.com).

## *In vivo* co-immunoprecipitation (co-IP) assays

For *in vivo* co-immunoprecipitation (co-IP) assays, *35S:HEC2-GFP*, *35S:mHEC2-GFP* and *35S:PIF4-myc* were crossed for each combination and double transgenic plants expressing both proteins were selected. Immunoprecipitation was conducted using 50 seedlings for each sample with Dynabeads protein A and anti-myc (Abcam, Cambridge, MA). Western blots using anti-myc (Cell Signaling Technologies, Danvers, MA) or anti-GFP antibody (Abcam, Cambridge, MA) were used to detect the proteins.

## *In Vitro* DNA pull-down assay using biotinylated DNA

The DNA pull-down assay was conducted as previously described with some modifications [56]. The biotin-labeled primer pair amplifying the *PIF4* G-box region is listed in S1 Table as previously described [49]. One microgram of biotin-labeled DNA from *PIF4* G-box region was used for binding assay with 100 ng of GST, GST-PIF4, and GST-HEC2 as previously expressed and purified from *E. coli* [28,38]. After 2hr of incubation for protein dimerization followed by DNA binding, streptavidin agarose beads (Catalog # S1420S; New England Biolabs, Ipswich, MA) were added and incubated for additional 30 min at 4˚C. Beads were thoroughly washed and boiled in SDS loading buffer. Western blots using anti-GST HRP (Catalog # RPN1236; GE healthcare, Pittsburgh, PA) was used to detect the proteins.

## Chromatin Immunoprecipitation (ChIP) assays

ChIP assays were performed as previously described [57]. Six-day-old seedlings of 35S:PIF4-myc in Col-0 or *hec1hec2* background were transferred to 22˚C or 28˚C for 24 hr in continuous white light and harvested. Sonication of chromatin pellet was performed using Branson digital sonifier (Emerson, St. Louis, MO). ChIP grade anti-Myc antibody (9B10, Abcam, Cambridge, MA) coupled to dynabeads were used for immunoprecipitation. Finally, PCR purification kit (Qiagen, Valencia, CA) was used for DNA purification. Samples without IP were used

as input DNA. Enrichment (% of input) was calculated from each sample relative to their corresponding input.

## Supporting information

**S1 Table. Primers used in this study.**
(PDF)

**S1 Fig. The expression of *HEC1* and *HEC2* is upregulated at high ambient temperature.** RNA-seq data show transcription level of *HEC1* and *HEC2* in WT comparing normal and high ambient temperature. Numbers in the bar graph indicate p-value. Sequence data were obtained from publicly available GEO web site under accession number GSE142354.
(TIFF)

**S2 Fig. Relative transcript level of *HEC2-GFP*.** RT-qPCR was performed to detect *HEC2-GFP* transcript level. Samples were from *35S:HEC2-GFP* whole seedling grown for 5 days in 22˚C and transferred to 22˚C or 28˚C for 4 hours. Three biological replicates were used in this study. Relative gene expression levels were normalized using expression levels of *ACT7*. n.s. stands for not significant according to Student's t-test (P<0.05).
(TIFF)

**S3 Fig. HECs regulate petiole elongation and flowering time at high ambient temperature.** (A) Box plot shows the petiole lengths of genotypes indicated. Seedlings were grown for two days in continuous white light at 22˚C and then either kept at 22˚C or transferred to 28˚C for additional 5 days. More than 10 seedlings were measured. The letters a-c indicate statistically significant differences based on one-way ANOVA analysis with Tukey's HSD test. Tukey's box plot was used with median as a center value. (B) Box plot shows the leaf number for bolting under long day conditions (16L:8D). Seedlings were grown for two days in continuous white light at 22˚C and then either kept at 22˚C or transferred to 28˚C until bolting. More than 10 seedlings were measured. The letters a-b indicate statistically significant differences based on one-way ANOVA analysis with Tukey's HSD test. Tukey's box plot was used with median as a center value.
(TIFF)

**S4 Fig. PIF family members contribute to thermomorphogenesis.** (A) Photograph shows seedling phenotypes of *PIF*s-overexpression lines and *phyB-9* at normal and high ambient temperature. Seedlings were grown for two days in continuous white light at 22˚C and then either kept at 22˚C or transferred to 28˚C for additional four days before being photographed. The scale bar represents 5 mm. (B) Box plot shows the hypocotyl lengths of seedlings described in (A). More than 10 seedlings were measured for each experiment and was repeated 3 times. The letters a-c indicate statistically significant differences between means of hypocotyl lengths (P<0.05) based on one-way ANOVA analysis with Tukey's HSD test. Tukey's box plot was used with median as a center value. (C, E) Western blots show the level of PIFs (C) and phyB (E) in Wild-type and various *PIF* overexpression lines. Seedlings were grown for 5 days at 22˚C and either kept at 22˚C or transferred to 28˚C for 4 hours. Coomassie staining or anti-RPT5 was used as a control. Red number indicates the quantitation value from anti-phyB or anti-myc detection divided by the control. (D and F) Dot plots show the relative amount of PIF4 (D, n = 3) or phyB (F, n = 3). Asterisks indicate statistically significant difference using Student's t-test; $^*$p < 0.05 and $^{**}$p < 0.01.
(TIFF)

**S5 Fig. Transcript levels of *PHYB* and *PIFs* in *PIFs*-overexpression lines.** RT-qPCR was performed to detect tagged-*PIF*s and *PHYB* transcript levels. Samples were from WT, *35S:TAP--PIF1*, *PIF3-myc*, *PIF4-myc* and *PIF5-myc* seedlings grown for 5 days at 22˚C and transferred to 22˚C or 28˚C for 4 hours. Three biological replicates were used in this study. Relative gene expression levels were normalized using the expression level of *ACT7*. n.s. stands for not significant according to Student's t-test (P<0.05).
(TIFF)

**S6 Fig. The expression of four major *PIFs* is upregulated at high ambient temperature.** RT-qPCR was performed to detect the transcript levels of four major *PIF*s (*PIF1*, *PIF3*, *PIF4* and *PIF5*). Samples were from WT seedling grown for 5 days at 22˚C and transferred to 22˚C or 28˚C for 4 hours. Three biological replicates were used in this study. Relative gene expression levels were normalized using the expression level of *ACT7*. Asterisks indicate statistically significant difference using Student's t-test; $^*$p < 0.05 and $^{**}$p < 0.01.
(TIFF)

**S7 Fig. PIFs are epistatic to HECs in regulating petiole length and flowering time.** (A) Box plot shows the petiole lengths of genotypes indicated. Seedlings were grown for two days in continuous white light at 22˚C and then either kept at 22˚C or transferred to 28˚C for additional 5 days. More than 10 seedlings were measured. The letters a-c indicate statistically significant differences based on one-way ANOVA analysis with Tukey's HSD test. Tukey's box plot was used with median as a center value. (B) Box plot shows the leaf number for bolting under long day conditions (16L:8D). Seedlings were grown for two days in continuous white light at 22˚C and then either kept at 22˚C or transferred to 28˚C until bolting. More than 10 seedlings were measured. The letters a-b indicate statistically significant differences based on one-way ANOVA analysis with Tukey's HSD test. Tukey's box plot was used with median as a center value.
(TIFF)

**S8 Fig. Gene Ontology (GO) analysis of the temperature-dependent 735 genes.** Gene Ontology (GO) analysis of temperature-dependent 735 genes that are common among three genotypes. Six-day-old white light-grown seedlings were transferred to 22˚C or 28˚C for additional 24 hours and total RNA was extracted from three biological replicates for RNA-seq analyses.
(TIFF)

**S9 Fig. The expression patterns of *HEC*- and *PIF*-dependent genes are opposite.** (A-B) Hierarchical clustering displaying 844 (638+206) *HEC*-dependent DEGs (A) and 1067 (638+429) *PIF*-dependent DEGs (B) shows distinct pattern in *hec1hec2* and *pifQ* mutant at high ambient temperature.
(TIFF)

**S10 Fig. Mutated HEC does not affect flowering time at high ambient temperature.** (A) Box plot shows the leaf number for bolting (16L:8D). More than 10 seedlings were measured. n.s. stands for not significant based on one-way ANOVA analysis with Tukey's HSD test. Tukey's box plot was used with median as a center value.
(TIFF)

**S11 Fig. HEC-PIF4 heterodimerization inhibits 26S proteasome-mediated degradation of PIF4.** (A) Western blot shows the level of PIF4-myc from whole seedlings of 35S:PIF4-myc in either *hec1hec2* or wild-type backgrounds. Seedlings were grown for 5 days at 22˚C and either kept at 22˚C or transferred to 28˚C for 4 hours. B stands for bortezomib treatment. Coomassie staining was used as a control. Red number indicates the quantitation value from anti-myc

divided by the control.
(TIFF)

**S12 Fig. HEC2 inhibits PIF4 binding to DNA *in vitro*.** (Upper panel) Immunoblot shows the amount of GST only (as a control), GST-PIF4, and GST-HEC2 and their combinations as input. All GST-fusion proteins were expressed and purified from E. coli and detected using anti-GST antibody. (Lower panel) Immunoblot shows the amount of GST-PIF4 bound to the DNA. Biotin labeled *PIF4* G-box promoter region was precipitated using streptavidin beads after pre-binding with combinations of proteins as indicated by + and/or -.
(TIFF)

**S13 Fig. The expression levels of *PIFs* in WT and *hec1hec2* at high ambient temperature.** RNA-seq data show transcription level of *PIF1*, *PIF3*, *PIF4*, *PIF5*, and *PIF7* in WT and *hec1hec2* background comparing normal and high ambient temperature. *h12* indicates *hec1 hec2* mutant. Numbers in the bar graph indicate p-value.
(TIFF)

## Acknowledgments

We thank Dr. Inyup Paik for helpful discussion throughout this project, members of the Huq laboratory for critical reading of the manuscript, and Dr. Peter Quail for sharing anti-phyB antibody and *phyB pifQ* seeds. The authors acknowledge the Texas Advanced Computing Center (TACC) at The University of Texas at Austin for providing High Performance Computing, visualization, and database resources that have contributed to the research results reported in this paper.

## Author Contributions

**Conceptualization:** Sanghwa Lee, Enamul Huq.

**Data curation:** Sanghwa Lee.

**Formal analysis:** Sanghwa Lee, Enamul Huq.

**Funding acquisition:** Enamul Huq.

**Investigation:** Sanghwa Lee.

**Methodology:** Sanghwa Lee, Ling Zhu.

**Project administration:** Sanghwa Lee, Enamul Huq.

**Resources:** Sanghwa Lee, Ling Zhu.

**Software:** Sanghwa Lee.

**Supervision:** Enamul Huq.

**Validation:** Sanghwa Lee.

**Visualization:** Sanghwa Lee.

**Writing – original draft:** Sanghwa Lee.

**Writing – review & editing:** Sanghwa Lee, Enamul Huq.

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
