## [Decision Letter · Decision Letter 0]

1 Mar 2021

Dear Dr Huq,

Thank you very much for submitting your Research Article entitled 'An autoregulatory negative feedback loop controls thermomorphogenesis in Arabidopsis' to PLOS Genetics.

The manuscript was fully evaluated at the editorial level and by independent peer reviewers. All reviewers are in support of this manuscript and highlight its significant interest to the field. In light of the level of interest in the manuscript, we would be willing to consider a resubmission that addresses the points raised by the reviewers. Reviewer 2 in particular has made a detailed suggestion about the logical flow of the manuscript, which would make the study easier to follow and clarify the underlying biology. We therefore ask you to modify the manuscript according to the review recommendations. Your revisions should address the specific points made by each reviewer.

[LINK]

Yours sincerely,

Philip Anthony Wigge, Ph.D

Guest Editor

PLOS Genetics

Claudia Köhler

Section Editor: Plant Genetics

PLOS Genetics

Reviewer's Responses to Questions

**Comments to the Authors:**

Reviewer #1: The manuscript by Lee et al. reports the regulatory role of HEC1 and HEC2 in thermomorphogenesis. The authors first demonstrate the involvement of HEC1/2 in thermomorphogenesis by showing that high ambient temperature induced HEC1/2 gene expression level and HEC2 protein stability, and that hec1/2 mutants and HEC2-ox lines displayed opposite thermomorphogenic phenotypes. They then show that all four PIFs (PIF1/3/4/5) contribute to promote thermomorphogenesis, and that pifQ is epistatic to hec1/2 by genetic analysis. Moreover, the authors show that HECs could interact and form heterodimers with PIF4, thus HECs could inhibit PIF binding to their target gene promoters. Consequently, HECs and PIFs oppositely control the expression of many high temperature-regulated genes, thus forming a negative feedback loop that fine-tunes plant growth under high ambient temperature.

Generally, this manuscript contains data that are both interesting and important, and would contribute to a better understanding of high ambient temperature-induced architecture changes of sessile plants. Here are my suggestions to improve this manuscript.

1) Abstract, “PIF4-HEC forms an autoregulatory composite negative feedback loop”: please rephrase.

2) Introduction, “Furthermore, several factors have been shown to control either the PIF4 abundance and/or activity to regulate thermomorphogenesis”: it was recently shown that the cold response regulator CBF1 promotes PIF4 abundance at ambient temperatures, and that MYB30 promotes PIF4 protein re-accumulation under prolonged light irradiation (EMBO J, 39:e103630; Plant Cell, 32:2196-2215). These recent reports are relevant to this statement.

3) It is important to show that all four PIFs contribute to promote thermomorphogenesis. In support of this claim, I would prefer to show the data of pif mutants instead of PIF-OE lines in the main figure, if only one could be included as a main figure. In this sense, I would suggest the authors to exchange Figure 2 with Figure S4.

4) Since the authors show that all four PIFs contribute to promote thermomorphogenesis, I would suggest the authors to make this story bigger, i.e., HECs not only interacts with and regulates the activity of PIF4, but also do the same with other PIFs. In this sense, the authors could detect the protein levels of PIF1/3/5 in Figure 1E, and could examine the expression levels of PIF1/3/5 in Figure 6B.

5) Figure 1E shows that both PIF4 and phyB protein levels are decreased in hec1 hec2 mutants at 28 degree. How could HEC1/2 promote PIF4 and phyB stability simultaneously at 28 degree? The authors may do some tests, at least make some discussion.

6) The authors may do some assays, such EMSAs, to provide direct evidence showing that HEC could inhibit PIF4 binding to its target gene promoter.

7) The model in Figure 6D should be improved to better illustrate the findings. The HEC inhibition of PIF4 is confusing because HECs promote PIF4 protein stability at high temperature.

Reviewer #2: The manuscript by Lee et al. shows that helix-loop-helix transcriptional regulators HECATE 1 (HEC1) and HEC2 inhibit thermomorphogenesis by physically interacting with PIF4 transcription factor and reducing its binding to PIF4 target genes. The carefully designed experiments are statistically solid and provide the necessary support to the conclusions. The work provides insight into our understanding of the mechanisms involved in thermomorphogenesis in plants.

My major concern is that the logic behind the general structure of the text is no easy to follow. I do not refer here to the wording itself but to the order in which the information is presented. Perhaps the authors are describing their findings in the order in which they did the experiments. It took me a while to find the main message. Therefore, I propose substantial cut and paste and some rearrangement of the figures as follows:

The first issue is that HECs increase their abundance in response to warmth and repress thermomorphogenesis. Therefore, Fig. 1 should contain parts A-D, leaving E and F for a later stage.

The second point of the manuscript is that the action of HECs requires PIFs. I know that the observation that all PIFs contribute to thermomorphogenesis is interesting in itself but it should not complicate the message. Therefore, I would merge Figs 2 and 3A-B into Fig. 2. Then, it makes sense if the authors are using the pifQ mutant background to demonstrate that the action of HECs requires PIFQ, to provide the information that all the PIFQ members contribute to thermomorphogenesis. By passing, since the authors have produced hec1 hec2 pifQ, they are likely to have hec1 hec2 pif4 in the segregating population. Since the rest of the paper deals with PIF4 it would be interesting to include the latter mutant (although I would not consider that a must do).

I would use current Fig. 5A-E as the third display (Fig. 3) because the convergence of the hec1 hec2 and pifQ transcriptomes further supports the conclusion reached by the genetic experiments in Fig. 2.

The fourth point is that although HECs affect PIF4 abundance these effects do not fully account for the requirement of PIFs for HECs action. Therefore, I would fuse PIF4 data from Fig. 1F-G plus Fig. 3C-D (PIFOX data) into Fig. 4.

The fifth point is that PIF4-HEC2 heterodimerization is crucial for HECs function in thermomorphogenesis and HECs reduce PIF4 binding to its targets. Therefore, I would use Fig. 4 plus Fig. 5G-H in Fig. 5.

The sixth point is the autoregulatory negative feedback loop (Fig. 6 as it is).

The subtitle “phyB and PIFs are oppositely regulated at high ambient temperature” deviates from the main stream and is confusing. It could be placed at the end, after the sequence described above.

Minor concerns

There are several known positive and negative regulators of thermomorphogenesis. Do expression patterns or other features of the HEC system tell us when (e.g. time of day, growth conditions) or where (organs) HECs could be important to reduce thermomorphogenesis?

Introduction: “Analyses of the four major PHYTOCHROME INTERACTING FACTORs (PIF1, PIF3, PIF4 and PIF5) mutants”. The “four major” would exclude PIF7 from the major PIFs, which is actually misleading.

Introduction: “while PIFs are acting positively downstream of phyB in attenuating thermormoprhogenesis”. I do not think “attenuating” is correct here.

Introduction: “An intriguing finding in our study is that PIF4 appears to regulate its own expression by directly binding to its own promoter”. Please, check carefully, I think that this had already been shown in the literature.

Figure S4. PIFs additively rgeulate thermomorphogenesis. It should be “regulate”

Reviewer #3: This manuscript describes the functional involvement of HEC1 and HEC2 in thermo responses. Their transcription and accumulation are enhanced by a higher ambient temperature. Overexpression of HEC2 attenuates thermo-induced hypocotyl elongation but enhances PIF4 stability or accumulation. hec1 hec2 double mutant shows enhanced thermo responses, and PIF4 is epistatic to HECs. HEC2 physically interacts with PIF4. Their interaction may inhibit 26S proteosome-mediated degradation of PIF4 but on the other hand may insulate PIF4 from binding to it target sites. PIF4 (or other PIFs) directly up-regulate the expression of HECs and HECs in turn inhibit the expression of PIF4. All discoveries are summarized in Figure 6D.

Major

1) In the fourth paragraph of the introduction, you discussed various ways that regulate the stability or activity of PIF4, and one particular case is mentioned for TOC1. There is a publication reflecting the involvement by the CCA1/LHY clock components that directly bind to the PIF4 promoter and activate its expression and likely accumulation (Sun et al., 2019, NC).

2) For Figures 5H and 6C, it would be helpful to audients by showing the promoter regions for ChIP-PCR analysis in a simple diagram.

Minor

1) In Figure S1 legend, add the reference number for your citation of Lee et al.

2) In Figure 2 legend, revise the description for C through F from WT, 35S:HEC2-GFP and hec12 mutant to wild type and various PIF overexpression lines.

3) In line 181, cite Figure 3A.

4) In Figure 3 legend, revise line 4 from as described in (B) to as described in (A).

**Have all data underlying the figures and results presented in the manuscript been provided?**

Reviewer #1: None

Reviewer #2: Yes

Reviewer #3: Yes

PLOS authors have the option to publish the peer review history of their article (what does this mean?). If published, this will include your full peer review and any attached files.

Reviewer #1: No

Reviewer #2: No

Reviewer #3: No

---

## [Editor Report · Decision Letter 1]

11 May 2021

Dear Dr Huq,

We are pleased to inform you that your manuscript entitled "An autoregulatory negative feedback loop controls thermomorphogenesis in Arabidopsis" has been editorially accepted for publication in PLOS Genetics. Congratulations!

Yours sincerely,

Philip Anthony Wigge, Ph.D

Guest Editor

PLOS Genetics

Claudia Köhler

Section Editor: Plant Genetics

PLOS Genetics

**Data Deposition**

http://datadryad.org/submit?journalID=pgenetics&manu=PGENETICS-D-21-00104R1

**Press Queries**

---

## [Editor Report · Acceptance letter]

24 May 2021

PGENETICS-D-21-00104R1 

An autoregulatory negative feedback loop controls thermomorphogenesis in Arabidopsis 

Dear Dr Huq, 

We are pleased to inform you that your manuscript entitled "An autoregulatory negative feedback loop controls thermomorphogenesis in Arabidopsis" has been formally accepted for publication in PLOS Genetics! Your manuscript is now with our production department and you will be notified of the publication date in due course.

With kind regards,

Katalin Szabo

PLOS Genetics

On behalf of:
